# FARFusion V2: A Geometry-based Radar-Camera Fusion Method on the Ground for Roadside Far-Range 3D Object Detection

Yao Li
zkdly@mail.ustc.edu.cn
University of Science and Technology
of China
Hefei, China

Jiajun Deng
jiajun.deng@sydney.edu.au
The University of Sydney
Sydney, Australia

Yuxuan Xiao
xiaoyx@mail.ustc.edu.cn
University of Science and Technology
of China
Hefei, China

Yingjie Wang
yingjiewang@mail.ustc.edu.cn
University of Science and Technology
of China
Hefei, China

Xiaomeng Chu
cxmeng@mail.ustc.edu.cn
University of Science and Technology
of China
Hefei, China

Jianmin Ji
jianmin@ustc.edu.cn
University of Science and Technology
of China
Hefei, China

Yanyong Zhang*
yanyongz@ustc.edu.cn
University of Science and Technology
of China
Hefei, China

## Abstract

Fusing the data of millimeter-wave Radar sensors and high-definition cameras has emerged as a viable approach to achieving precise 3D object detection for roadside traffic surveillance. For roadside perception systems, earlier studies have pointed out that it is better to perform the fusion on the 2D image plane than on the BEV plane (which is popular for on-car perception systems), especially when the perception range is large (*e.g.*, $> 150m$). Image-plane fusion requires critical transformations, like perspective projection from the Radar's BEV to the camera's 2D plane and reverse IPM. However, real-world issues like uneven terrain and sensor movement degrade these transformations' precision, impacting fusion effectiveness. To alleviate these issues, we propose a geometry-based Radar-camera fusion method on the ground, namely FARFusion V2. Specifically, we extend the ground-plane assumption in FARFusion [20] to support arbitrary shapes by formulating the ground height as an implicit representation based on geometric transformations. By incorporating the ground information, we can enhance Radar data with target height measurements. Consequently, we can thus project the enhanced Radar data onto the 2D plane to obtain more accurate depth information, thereby assisting the IPM process. A real-time parameterized transformation parameters estimation module is further introduced to refine the view transformation processes. Moreover, considering various measurement noises across these two sensors, we introduce an uncertainty-based depth fusion strategy into the 2D fusion process to maximize the probability of obtaining the optimal depth value. Extensive experiments are conducted on our collected roadside OWL benchmark, demonstrating the excellent localization capacity of FARFusion V2 in far-range scenarios. Our method achieves an average location accuracy of 0.771m when we extend the detection range up to 500m.

## CCS Concepts

• **Computing methodologies → Object detection**.

## Keywords

3D Object detection, Sensor fusion, Intelligent Transportation System

**ACM Reference Format:**
Yao Li, Jiajun Deng, Yuxuan Xiao, Yingjie Wang, Xiaomeng Chu, Jianmin Ji, and Yanyong Zhang. 2024. FARFusion V2: A Geometry-based Radar-Camera Fusion Method on the Ground for Roadside Far-Range 3D Object Detection. In *Proceedings of the 32nd ACM International Conference on Multimedia (MM '24), October 28–November 1, 2024, Melbourne, VIC, Australia.* ACM, New York, NY, USA, 10 pages. https://doi.org/10.1145/3664647.3681128

---

*Corresponding author.

## 1 Introduction

Far-range 3D object detection and tracking are crucial for roadside perception in intelligent transportation systems, as highlighted in recent studies [4, 7, 33]. These technologies can significantly enhance roadside-assisted autonomous driving and provide early warnings of potential traffic incidents [2, 13, 29, 41, 46]. To ensure precise vehicle location at large distances, deploying both millimeter wave radar (referred to as Radar in this paper) and high-definition (HD) cameras for joint perception has become an emerging solution in roadside scenes [8]. Feature-level Radar and camera fusion

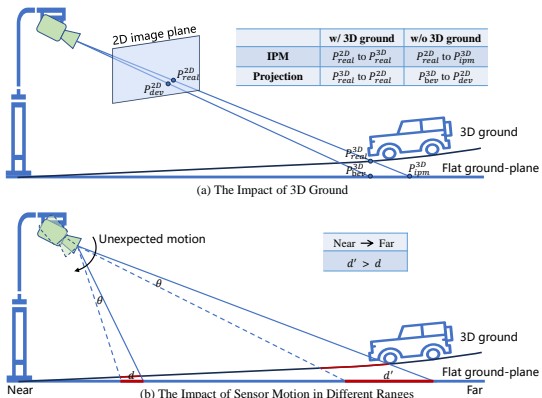

| | w/ 3D ground | w/o 3D ground |
|---|---|---|
| **IPM** | $P_{real}^{2D}$ to $P_{real}^{3D}$ | $P_{real}^{2D}$ to $P_{ipm}^{3D}$ |
| **Projection** | $P_{real}^{3D}$ to $P_{real}^{2D}$ | $P_{bev}^{3D}$ to $P_{dev}^{2D}$ |

(a) The Impact of 3D Ground

| Near → Far |
|---|
| $d' > d$ |

(b) The Impact of Sensor Motion in Different Ranges

**Figure 1: (a) With the 3D ground, we can project the 3D point $P_{real}^{3D}$ to the $P_{real}^{2D}$ and also can inversely map the 2D point $P_{real}^{2D}$ to the $P_{real}^{3D}$ via IPM. Differently, with the flat ground plane assumption, we can only project the $P_{bev}^{3D}$ to the deviated position $P_{dev}^{2D}$ on the 2D plane and inversely map the $P_{real}^{2D}$ to the deviated position $P_{ipm}^{3D}$ in the 3D space via IPM. (b) If the camera has an unexpected movement with a pitch angle $\theta$, there will be a larger deviation $d'$ at far ranges.**

approaches [17, 34, 35, 43, 44] have achieved high performance. However, they usually require extensive annotated data [32] that is hard to obtain in far-range scenes. Therefore, target-level fusion methods are primarily used in practical systems.

Fusion on the image plane first is more reliable for target-level fusion as illustrated in FARFusion [20], especially for far-range roadside scenes (*e.g.,* > 150m). Specifically, lifting 2D targets into 3D space directly suffers from larger location errors, causing incorrect association with Radar targets on the BEV plane. Thus, FARFusion first projects Radar targets to the 2D plane to associate with 2D targets, then maps 2D targets to the BEV plane using inverse perspective mapping (IPM) with Radar depth information. FARFusion also performs the fusion on the ground. This fusion strategy requires critical view transformations between Radar targets in 3D space and camera targets on the 2D plane. However, mapping from 2D to 3D space is an ill-posed problem [5, 21, 26] since a point on the image plane corresponds to all collinear 3D points on the ray from the optical center. Inspired by Monoground [26], FARFusion represents vehicles as Radar points or camera 2D points on the ground. On the BEV plane, the Radar coordinate system is parallel to the ground plane because it is aligned to the UTM coordinate system in initial calibration [20]. On the 2D image plane, FARFusion models each object as a point at the contact area between the vehicle bottom and the ground, creating a one-to-one correspondence for the 2D-3D mapping. These strategies enable FARFusion to achieve accurate localization in far-range scenarios.

However, during the fusion process, FARFusion assumes that the road is a flat plane, which may not hold in real far-range scenarios. The uneven road in the real world will affect the precision of view transformations of both perspective projection and IPM processes shown in Fig. 1(a). Specifically, the commonly used Radar sensors lack height measurement, so we usually assign a fixed height to Radar data when projecting Radar points from the BEV plane to

the 2D image plane. This will result in inaccurate projection depth values and positions on uneven roads. Meanwhile, IPM also relies on the planar assumption [40, 42], which causes deviation between IPM points and actual positions on uneven roads. Additionally, FARFusion also assumes the roadside sensors are fixed in a time interval. Unfortunately, real-time unexpected sensor motions still exist due to the windy weather, causing inaccurate transformation between 2D and 3D spaces especially at larger distances shown in Fig. 1(b). Here we only show the motion in the pitch angle, as this direction has the most significant impact on depth estimation.

In this work, we propose FARFusion V2 to mitigate the above issues for practical far-range roadside perception. Our framework has two key designs – implicit ground height learning and real-time transformation parameters estimation. Firstly, considering the ground may have arbitrary shapes, we represent ground height as a learnable function. We also assume there is a virtual ground plane. Then we utilize a customized loss function to train the function on both 2D and BEV planes. Specifically, we integrate prior geometric relations [10, 20] between real and virtual ground planes into the training process. With the ground function, we can query the ground height at any location, which allows us to attach height information to the Radar data. Consequently, we can thus project the enhanced Radar points onto the 2D plane to obtain more accurate depth information, thereby assisting the IPM process. Secondly, roadside camera images contain rich background information, including the camera's relative pose to the ground [18, 23]. Therefore, we directly regress the transformation parameters in real time to refine the view transformations. To facilitate the learning of transformation parameters, we enable the network to regress the residuals of the parameterized homography matrix between 2D and BEV planes. These key designs in FARFusion V2 can boost the image-plane fusion pipeline proposed by FARFusion for far-range scenes. Moreover, considering various observation noises across these two sensors, we introduce an uncertainty-based depth fusion strategy into the image-plane fusion. This strategy aims to maximize the probability of obtaining the optimal depth value. Meanwhile, due to the absence of direct labels for uncertainty, we utilize a surrogate loss function to train the network for uncertainty estimation.

In summary, we make the following contributions:

- We propose a geometry-based Radar-camera fusion method for far-range object detection on the ground, namely FARFusion V2. Our approach achieves excellent localization by integrating the ground information and a real-time transformation parameters estimation module in target-level fusion.
- We also introduce an uncertainty-based depth fusion strategy into the fusion process to maximize the probability of obtaining the optimal depth value.
- Extensive experiments are conducted on our roadside OWL testbed on an urban expressway. Results show that our method can improve $AP_{BEV}$ by absolutely 5.4%. Significantly, the approach can achieve $0.771m$ average location accuracy when the detection range is extended up to $500m$.

## 2 Related Work

Radar and camera have been widely deployed in autonomous driving (AD) and roadside scenes thanks to their low costs [19].

**Radar and camera fusion for autonomous driving.** Feature-level [22] and target-level fusion methods are the primary methods for integrating Radar and camera data in autonomous driving. We first introduce the feature-level methods, CenterFusion [24] first predicts 2D candidate positions of the targets and then utilizes the roi-based fusion strategy to associate them with Radar pillar features in 3D space via inverse perspective transformation. Finally, it refines the associated features using a refinement module. Cramnet [14] also applies such an association strategy to map the foreground pixels to 3D for point-wise fusion with foreground Radar points. Different from the roi-based fusion strategy, BEV-based fusion methods [43, 45] first lift the 2D image features to BEV features in 3D space and then fuse them with the Radar BEV features. In the target-level fusion methods [28, 31], the fusion is performed on the target-level sensor outputs. Most methods [3, 6, 9, 16, 25] are based on Bayesian theory, Kalman Filtering, Dempster-Shafer theory, etc. Target-level fusion has been widely employed in practical scenarios thanks to the low computing cost.

**Radar and camera fusion for roadside perception.** Nowadays, more research is dedicated to roadside dataset building [11, 37–39], sensor calibration [8, 27] and object location [1]. However, there has been relatively little work on combining roadside radar and camera data. [1] utilizes a BEV-based method to fuse Radar and camera targets, it first maps the 2D points to BEV and associates two targets on the BEV plane, then uses a Gaussian mixture probability hypothesis density (GM-PHD) filter to integrate these two modalities. [8] also utilizes the BEV-based method to achieve spatiotemporal synchronization between Radar and camera coordinate systems. It first maps the 2D points to BEV and then employs the matched trajectory points to optimize the calibration parameter. [30] fuses roadside Radar and camera targets using the Kalman filter in the tracking stage. These studies primarily focus on near-range perception (less than 250 meters). Differently, FARFusion [20] proposes a far-range Radar and camera fusion method for 3D object detection, which first associates Radar-camera targets on the 2D plane and refines the transformation parameter on both 2D and BEV planes.

In summary, most Radar and camera fusion methods for AD primarily use feature-level fusion strategies. Roadside Radar and camera fusion methods mainly utilize the target-level fusion strategy due to the lack of accessible open datasets in near range (<250m). By contrast, our FARFusion V2 focuses on roadside far-range ($150 - 500m$) perception, taking into account the key impacts of rough ground and sensor movements on the high-accuracy location.

## 3 The FARFusion V2 Design

### 3.1 Preliminary and Overview

FARFusion V2 aims to enhance the precision of fusing Radar and camera targets in far-range roadside scenes. We first introduce the basic fusion pipeline designed for far-range scenarios in FARFusion [20]. FARFusion first projects the Radar-based target points (on the BEV plane) to the 2D image plane and then associates them with the camera-based object locations that are modeled as a CBM[1] point on each object. Subsequently, it maps the camera-based object locations to the BEV plane through IPM with the corresponding

---

[1]CBM represents the bottom midpoint of the 2D bounding box detected by a pre-trained 2D detector, which is a good proxy of the car's location.

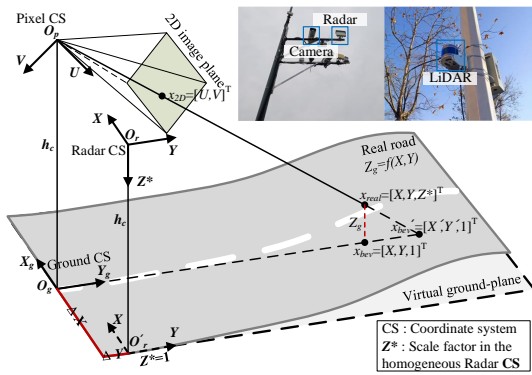

**Figure 2: The coordinate systems of the OWL testbed.**

Radar depth information. Finally, it engages a BEV tracking module to combine the outputs of the above 2D fusion module and generate target trajectories for traffic monitoring. Meanwhile, the fusion pipeline requires accurate transformations between Radar and pixel coordinate systems (CS). We follow FARFusion to represent the transformation parameters using a plane homography matrix $\mathbf{M}$ between Radar BEV and image planes. The coordinate systems are depicted in Fig. 2 on the OWL testbed deployed on an urban expressway with a detection area of $Y = 150 - 500m$ range.

In the FARFusion, the BEV plane (Radar CS) corresponds to the ground surface, which assumes the ground is a plane without relief. This assumption may be inconsistent with actual circumstances, especially in far-range scenes, which has adverse effects on the projection process of Radar points and IPM process of camera CBM points. Therefore, we have revised this assumption to allow the ground to have arbitrary shapes in FARFusion V2. Moreover, FARFusion assumes that the camera is mounted in a pole without movements in a time window. However, we have observed the camera has little shake due to the windy weather, which influences the accuracy of the pre-calibrated $\mathbf{M}$. Considering roadside camera images include the inherent pose information relative to the ground, we refine the $\mathbf{M}$ from each image in real time in FARFusion V2.

Specifically, in our FARFusion V2, given a pre-calibrated $\mathbf{M}$ and a set $X = \{x_{bev}^i, x_{2D}^i | i = 1...N\}$ of $N$ pair detection points of vehicles for every frame, with BEV object coordinates $x_{bev}^i \in \mathbb{R}^2$ in Radar CS and 2D pixel coordinates $x_{2D}^i \in \mathbb{R}^2$, our goal is to estimate ground height and refine $\mathbf{M}$ in real time, then enhance the target-level Radar and camera fusion for far-range scenes. Here, considering it's usually hard to obtain the 3D ground truth of targets in the whole range of $Y = 150 - 550m$ in the Radar CS, we take fusion points $(X_f, Y_f)$ as $x_{bev}^i$ and 2D CBM points $(U_{cam}, V_{cam})$ as $x_{2D}^i$. The fusion pipeline in FARFusion provides these two types of points and their matching relations. Note that our FARFusion V2 attempts to construct data-driven surrogate training pipelines using prior geometric transformations and the outputs of FARFusion, instead of taking the outputs as the direct supervision. To this end, with the $X$ and $\mathbf{M}$, our FARFusion V2 proposes three key designs: (1) A ground height network that uses a geometry-based training pipeline. (2) A real-time parameterized homography matrix estimation module. (3) An uncertainty-based target-level Radar and camera fusion module. The framework is depicted in Fig. 3. Below we will introduce every

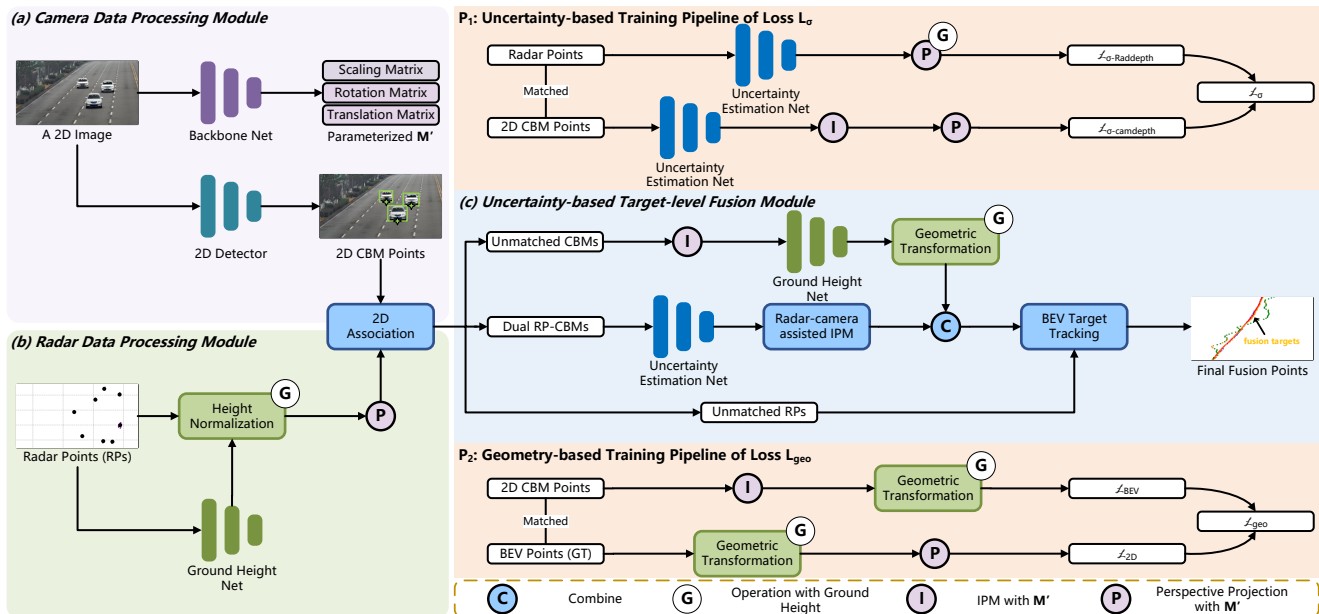

Figure 3: The overview of our FARFusion V2. FARFusion V2 mainly contains three modules: (a) The camera data processing module, which employs a 2D detector to detect 2D CBM (camera-object bottom midpoints of the 2D bounding box) points and a 2D image backbone to estimate the parameterized transformation parameter $\mathbf{M}'$. (b) The Radar data processing module integrates the height information into Radar points with a ground height network and then projects Radar points to the 2D plane. (c) The uncertainty-based target-level fusion module. This module associates Radar points with 2D CBM points on the 2D plane. Then it maps dual Radar-CBM points to 3D space with weighted depths calculated by the uncertainty-based networks. The whole framework is trained using the geometry-based loss function $L_{geo}$ and the uncertainty-based loss function $L_{\sigma}$.

module in more depth. To make it easily readable, we first give the homography transformation between Radar CS and pixel CS as:

$$Z\begin{pmatrix} U \\ V \\ 1 \end{pmatrix} = \mathbf{M}\begin{pmatrix} X \\ Y \\ 1 \end{pmatrix} = \begin{pmatrix} m_{11} & m_{12} & m_{13} \\ m_{21} & m_{22} & m_{23} \\ m_{31} & m_{32} & m_{33} \end{pmatrix}\begin{pmatrix} X \\ Y \\ 1 \end{pmatrix}. \qquad (1)$$

## 3.2 Geometry-based Ground Height Learning

We represent the ground height implicitly and construct a geometry-based training pipeline for ground height learning. We first assume there is a virtual ground plane shown in Fig. 2, and then we utilize the differences between predicted positions based on the hypothesis and real positions to train the ground height network. Meanwhile, $\mathbf{M}$ denotes the transformation between 2D and virtual BEV planes. During the initial calibration [20], the ground is assumed as a plane with a fixed ground height value to calibrate the $\mathbf{M}$.

Specifically, considering the requirements of practical deployment with arbitrary ground shapes, we first represent the ground height as follows:

$$Z_g = f(X, Y), \qquad (2)$$

where in our case the height function is implicitly parameterized as a Multi-Layer Perceptron (MLP).

Then given the pair fusion points $(X_f, Y_f)$ and 2D CBM points $(U_{cam}, V_{cam})$, we construct the loss function on both 2D and BEV virtual planes depicted in Fig. 3.$P_2$. Firstly, we start from the 2D CBM points $(U_{cam}, V_{cam})$. Based on the virtual ground-plane assumption,

we map the 2D CBM points to the BEV plane via the IPM process:

$$X_c' = \mathbf{H}_1 \cdot Z_c(U_{cam}, V_{cam}, 1), \qquad (3)$$

$$Y_c' = \mathbf{H}_2 \cdot Z_c(U_{cam}, V_{cam}, 1), \qquad (4)$$

where $\mathbf{H}$ is the inverse of $\mathbf{M}$, $\mathbf{H}_i$ denotes the i-th row of $\mathbf{H}$, and $Z_c = 1/(\mathbf{H}_3 \cdot (U_{cam}, V_{cam}, 1))$. Then with the Equ. 2, we can obtain the height $Z_g$ at the real position $(X_c, Y_c)$ from $Z_g = f(X_c', Y_c')$. Here the mapping relation between $Z_g$ and $(X_c', Y_c')$ is one-to-one as the intersection points of the single ray from camera optical center (with a height $h_c$) with the real ground and the virtual plane should be unique in the roadside scene. These points lie on a single ray shown in Fig. 2. Inspired by the Gen-LaneNet[10], we transfer the $(X_c', Y_c')$ on the virtual plane to the real position $(X_c, Y_c)$ as follows:

$$X_c = (1 - \frac{Z_g}{h_c})X_c' - \frac{Z_g}{h_c}\Delta X, \qquad (5)$$

$$Y_c = (1 - \frac{Z_g}{h_c})Y_c' - \frac{Z_g}{h_c}\Delta Y, \qquad (6)$$

where $\Delta X$ and $\Delta Y$ are the translation deviations between Radar CS and ground CS shown in Fig. 2, $h_c$ is the perpendicular distance from the camera's optical center to the ground. The variables $\Delta X$, $\Delta Y$ and $h_c$ are considered as learnable parameters. We consider the translation deviations because the geometry-based similar triangle relation ($\frac{X_c + \Delta X}{X_c' + \Delta X} = \frac{h_c - Z_g}{h_c}$) is only established with the ground CS while our BEV plane is in the Radar CS. Therefore we re-derive the

Equ. 5 and Equ. 6. Then we take the fusion points $(X_f, Y_f)$ as the reference and calculate the first loss function on the BEV plane as:

$$L_{BEV} = (X_c - X_f)^2 + (Y_c - Y_f)^2. \tag{7}$$

Secondly, we start from the fusion points $(X_f, Y_f)$ to calculate the re-projection errors on the 2D plane. In the training process, the matching relation of fusion points $(X_f, Y_f)$ and 2D CBM points $(U_{cam}, V_{cam})$ are known, therefore we can take the ground height $Z_g$ obtained in the calculation process of $L_{BEV}$ to transfer the $(X_f, Y_f)$ from real position to the position on the virtual ground plane. The transformation process is formulated as follows:

$$X'_f = \frac{h_c}{h_c - Z_g} X_f + \frac{Z_g}{h_c - Z_g} \Delta X, \tag{8}$$

$$Y'_f = \frac{h_c}{h_c - Z_g} Y_f + \frac{Z_g}{h_c - Z_g} \Delta Y. \tag{9}$$

Then we can project the $(X'_f, Y'_f)$ on the virtual plane to the 2D plane with the homography matrix $\mathbf{M}$ as:

$$U_f = \mathbf{M}_1 \cdot (X'_f, Y'_f, 1)/(\mathbf{M}_3 \cdot (X'_f, Y'_f, 1)), \tag{10}$$

$$V_f = \mathbf{M}_2 \cdot (X'_f, Y'_f, 1)/(\mathbf{M}_3 \cdot (X'_f, Y'_f, 1)). \tag{11}$$

Now we take the 2D CBM points $(U_{cam}, V_{cam})$ as reference and calculate the second loss function on the 2D plane as follows:

$$L_{2D} = (U_f - U_{cam})^2 + (V_f - V_{cam})^2. \tag{12}$$

The final geometric loss function can be calculated with all $N$ pair fusion and 2D CBM points by:

$$L_{geo} = \frac{1}{N} \sum_{i=1}^{N} (L_{2D} + L_{BEV})_i. \tag{13}$$

Inspired by the FARFusion [20], we apply the depth scaling strategy on the final geometric loss function due to the far-range detection requirements of $Y = 150 - 500m$ range. We also use the smooth $L_1$ loss on the $L_{2D}$ and $L_{BEV}$, which is robust to outliers and easier for ground height training.

## 3.3 Camera Data Processing Module

Our camera data processing module takes high-resolution images as inputs. As mentioned before, far-range object detection highly relies on the homography matrix $\mathbf{M}$ for view transformations between 2D and virtual BEV planes. In practical deployment, the sensors may have unexpected movements due to windy weather. Therefore we first exploit a parameterized homography matrix estimation method in our camera data processing module depicted in Fig. 3(a).
**Parameterized homography matrix estimation.** The roadside sensors are fixed and installed on the pole, any deviations in unexpected movements should be measured relative to a fixed reference parameter, *i.e.,* $\mathbf{M}$. We directly regress the residuals relative to the initial $\mathbf{M}$ for every image. The image includes rich background information, which is helpful for estimating the camera pose relative to the ground. Specifically, we employ a 2D image backbone to extract the down-sampled image feature map from every image. Then the image feature map is flattened to a 1-D global feature $M_I \in \mathbb{R}^{1 \times C}$. We transform the $M_I$ to a feature embedding $M'_I \in \mathbb{R}^{1 \times C'}$ for calibration using a MLP. The $M'_I$ is fed into different MLP layers to regress the residuals in parameterized representation relative to the

$\mathbf{M}$. We regress the scaling parameter $(\Delta s_x, \Delta s_y)$, rotation parameter $\Delta\theta$ and translation parameter $(\Delta t_x, \Delta t_y)$ separately. Finally, we derive the residual matrix $\Delta\mathbf{M}$ as follows:

$$\Delta\mathbf{M} = \begin{pmatrix} \Delta s_x & 0 & 0 \\ 0 & \Delta s_y & 0 \\ 0 & 0 & 1 \end{pmatrix} \begin{pmatrix} 1 & 0 & \Delta t_x \\ 0 & 1 & \Delta t_y \\ 0 & 0 & 1 \end{pmatrix} \begin{pmatrix} \cos\Delta\theta & -\sin\Delta\theta & 0 \\ \sin\Delta\theta & \cos\Delta\theta & 0 \\ 0 & 0 & 1 \end{pmatrix}. \tag{14}$$

Thus, we can refine $\mathbf{M}$ by matrix multiplication $\mathbf{M}' = \mathbf{M}\Delta\mathbf{M}$ for each frame. During training, we utilize these steps to transfer the $\mathbf{M}$ in the geometric transformation in Sec. 3.2 and use the loss function $L_{geo}$ in Equ. 13 to optimize the above residuals estimation network.
**2D CBM points detection.** We follow FARFusion [20] to detect the 2D CBM points using a pre-trained 2D detector.

## 3.4 Radar Data Processing Module

Radar can provide sparse point clouds and target-level results after clustering. We utilize target-level results that have more true positives as the inputs of the Radar data processing module, with each target represented by a point on the BEV plane. We first integrate height information into the Radar data. Given the Radar points $(X_r, Y_r)$ on the BEV plane, we approximately calculate the corresponding ground height $Z_g$ using height function $f$ in Equ. 2. This height value represents the Radar targets' height measurement, as our fusion process is performed on the ground. Then as shown in Fig. 2, we normalize $Z_g$ as $Z_g^{norm} = (h_c - Z_g)/h_c$. The normalization is necessary because we employ the homogeneous coordinate representation in the transformation between 2D and BEV planes. The virtual plane corresponds to the plane of $Z = 1$ in the Radar CS. Then we project the enhanced Radar points $(X_r, Y_r, Z_g^{norm})$ via perspective projection to calculate the projected depth $Z_{rad}$ and 2D coordinates $(U_{rad}, V_{rad})$ on the 2D plane as follows:

$$Z_{rad}\begin{pmatrix} U_{rad} \\ V_{rad} \\ 1 \end{pmatrix} = \mathbf{M}' \begin{pmatrix} X_r \\ Y_r \\ Z_g^{norm} \end{pmatrix} = \begin{pmatrix} m'_{11} & m'_{12} & m'_{13} \\ m'_{21} & m'_{22} & m'_{23} \\ m_{31} & m_{32} & m_{33} \end{pmatrix} \begin{pmatrix} X_r \\ Y_r \\ Z_g^{norm} \end{pmatrix}. \tag{15}$$

## 3.5 Uncertainty-based Target-level Radar and Camera Fusion

We next combine the Radar points and camera 2D CBM points through a 2-stage fusion pipeline on both 2D and BEV planes.

On the 2D plane, we perform an uncertainty-based depth fusion between 2D CBM points and Radar points depicted in Fig. 3(c). We first associate the Radar projection points with CBM points via the bipartite graph matching algorithm. After association, we have both matched point pairs and unmatched points. If a pair of points are matched, $(U_{cam}, V_{cam})$ and $(U_{rad}, V_{rad})$, we now regard the two points as a dual-modal point and map this point to the 3D space. For the matched CBM point $(U_{cam}, V_{cam})$, we have two types of depth values, *i.e.,* Radar projection depth $Z_{rad}$ and image depth $Z_c$. Here, we calculate $Z_c$ by the IPM process $Z_c = 1/\mathbf{H}'_3 \cdot (U_{cam}, V_{cam}, 1)$, where $\mathbf{H}'$ is the inverse of $\mathbf{M}'$. We assume that the two depths $Z_{rad}$ and $Z_c$ are independent because they are obtained from different methods. According to Bayes' theorem, we can derive the posterior probability of this 2D CBM point's real depth $D$ as below:

$$P(D|Z_c, Z_{rad}) \quad \propto \quad P(D|Z_{rad})P(D|Z_c). \tag{16}$$

 

Furthermore, we assume the observation noises of $Z_{rad}$ and $Z_c$ are subject to Gaussian distributions $N(0, \sigma_r^2)$ and $N(0, \sigma_c^2)$. According to the maximum of the posterior probability $P(D|Z_c, Z_{rad})$, we can obtain the following mathematical relation:

$$
\begin{aligned}
\max \quad & \log P(D|Z_c, Z_{rad}) \propto \max \log P(D|Z_{rad}) P(D|Z_c) \\
\propto \quad & \max \log \frac{1}{\sigma_r \sqrt{2\pi}} e^{-\frac{(D-Z_{rad})^2}{2\sigma_r^2}} + \log \frac{1}{\sigma_c \sqrt{2\pi}} e^{-\frac{(D-Z_c)^2}{2\sigma_c^2}} \\
\propto \quad & \min \frac{(D-Z_{rad})^2}{2\sigma_r^2} + \frac{(D-Z_c)^2}{2\sigma_c^2} + \log \sigma_r + \log \sigma_c. \quad (17)
\end{aligned}
$$

Thus, we can calculate the derivative of the final expression (denoted by $L(D)$) in Equ. 17 concerning $D$. By making $L(D)' = 0$, we can calculate the optimal $D$ as follows:

$$
D_{r+c} = \frac{Z_{rad}/\sigma_r^2 + Z_c/\sigma_c^2}{1/\sigma_r^2 + 1/\sigma_c^2}. \quad (18)
$$

Through the mathematical derivation, we thus obtain the optimal fusion depth value in Equ. 18. Considering the Radar and camera have different measurement noises in different ranges, we directly estimate the noises $\sigma_r, \sigma_c$ as the aleatoric uncertainties [15] of the $Z_{rad}$ and $Z_c$ using two MLP networks, i.e., $\sigma_r = MLP(X_r, Y_r)$, $\sigma_c = MLP(U_{cam}, V_{cam})$. Then we can map this dual-modal point to the real position in BEV with the coordinates $(X_{r+c}, Y_{r+c})$ by:

$$
X_{r+c} = \mathbf{H}_1' \cdot D_{r+c}(U_{cam}, V_{cam}, 1), \quad (19)
$$

$$
Y_{r+c} = \mathbf{H}_2' \cdot D_{r+c}(U_{cam}, V_{cam}, 1). \quad (20)
$$

As for the training process, we employ a surrogate loss function. According to the Equ. 17, we construct the loss function as follows:

$$
L_\sigma = \frac{(D-Z_{rad})^2}{2\sigma_r^2} + \frac{(D-Z_c)^2}{2\sigma_c^2} + \alpha \log \sigma_r + \beta \log \sigma_c, \quad (21)
$$

where $\alpha$ and $\beta$ are the hyperparameters and we take the fusion points as the reference to calculate the $D$ by $D = \mathbf{M}_3' \cdot (X_f, Y_f, Z_f^{norm})$. Note that the calculation of $D, Z_{rad}$ and $Z_c$ are detached from the computational graph in Sec. 3.2 and Sec. 3.3. $L_\sigma$ is only employed to train the two uncertainty estimation MLP networks. This training process is depicted in Fig. 3.$P_1$.

Additionally, for unmatched 2D CBM points, we utilize the IPM process to map the CBM points to the virtual BEV plane. Then we calculate the corresponding ground heights of the real positions via the height function in Equ. 2, and transfer the IPM points to the real positions using the Equ. 5 and Equ. 6. We also reserve the unmatched Radar points. We then feed all these outputs to the subsequent BEV target tracking module.

Finally, on the BEV plane, we adopt the common target-level fusion methods [20, 37], utilizing a Kalman-based tracking module to combine the outputs from the 2D depth fusion module.

## 4 Implementation and Evaluation

In this section, we introduce our implementation details and evaluation results.

### 4.1 Implementation Details

**OWL testbed and our benchmark.** As shown in Fig. 2, we have developed a testbed named OWL by deploying smart poles on an

**Table 1: Detection and location MAE results on validation set in $Y = 150 - 330m$ range.**

| Method | $\overline{AP}_{BEV}(\%) \uparrow$ | $AP_{BEV}(\%) \uparrow$ | | | | Loc. MAE$(m) \downarrow$ | | |
|---|---|---|---|---|---|---|---|---|
| | | $dis_{0.5}$ | $dis_{1.0}$ | $dis_{1.5}$ | $dis_{2.0}$ | Lat. | Long. | Euc. |
| *Radar Only* | 44.1 | 0.4 | 24.0 | 67.6 | 84.5 | 0.520 | 1.707 | 1.826 |
| *Image Only* | 16.6 | 2.0 | 9.5 | 21.6 | 33.5 | **0.126** | 1.526 | 1.541 |
| *robustBEVfus** [1] | 41.3 | 2.4 | 23.5 | 59.2 | 80.3 | 0.310 | 1.660 | 1.716 |
| FARFusion [20] | 63.9 | 15.8 | 60.9 | 85.6 | **93.3** | 0.142 | 1.217 | 1.239 |
| FARFusion V2 | **69.3** | **28.5** | **69.7** | **87.5** | 91.7 | 0.163 | 0.975 | **1.009** |

LiDAR provides the ground truth data of multiple cars in this range. *We follow [1] to exploit the same fusion method of the BEV association and "prediction-update" tracking pipeline as a comparison. Meanwhile, we use the Kalman filter and filter out the unmatched camera targets due to the large location errors of the IPM points in the tracking pipeline.

urban expressway with an $80km/h$ speed limit. Specifically, we deploy a long-range millimeter wave Radar, a HD monocular camera, a spinning 80-beam LiDAR on each pole, and two vehicle-mounted RTK-GPS systems. Here, LiDAR and RTK-GPS are deployed to provide ground truth data only for evaluation. LiDAR generates the data for vehicles in $Y = 150 - 330m$ range, while GPS provides the data for the two experimental vehicles in $Y = 330 - 500m$ range.

On this testbed, we establish a roadside benchmark for far-range scenarios. We follow [20] to perform spatiotemporal synchronization on our roadside dataset. Then We split the dataset into training (65%, first 65% frames per split) and validation splits (35%, other 35% frames per split) across three splits in different time intervals. Every split contains over 8000 frames and 39000 car samples.

**Metrics.** We use average precision (AP) and mean absolute error (MAE) of location to evaluate the final detection results in our benchmark. We assign the detection results to ground truths of vehicles based on the center distances between them, using different distance thresholds. We set the (0.5m, 1.0m) (i.e., $dis_{0.5}$), (1.0m, 2.0m) (i.e., $dis_{1.0}$), (1.5m, 3.0m) (i.e., $dis_{1.5}$) and (2.0m, 4.0m) (i.e., $dis_{2.0}$) thresholds in $(X, Y)$ axes (refer to Fig. 2) in the BEV's Radar coordinate system, because the length-width ratio of the vehicle is approximately $2 : 1$. We report location MAEs in the lateral (Lat.) $X$ axis, longitudinal (Long.) $Y$ axis and Euclidean distance (Euc.) from the bird's eye view with the threshold $dis_{1.5}$. We calculate the AP with location threshold values of $\{dis_{0.5}, dis_{1.0}, dis_{1.5}, dis_{2.0}\}$ and then average the values across different thresholds.

**Network structure.** The ground height and uncertainty estimation networks are parameterized as different MLPs with three layers and 32 channels of the hidden layer. In the parameterized homography matrix estimation module, we take the image with a resolution of $3840 \times 2160$ as input. Then we utilize the ResNet-18 [12] as the image backbone to extract the image feature for transformation parameters estimation. The hidden layers of different parameterized homography matrix estimation MLPs are configured with 256 channels each. We employ the pre-trained YOLOE [36] to detect the 2D bounding boxes of vehicles. We don't use the same image backbone for both calibration and object detection tasks because they separately focus on static and dynamic targets.

**Training details.** We decouple the training processes of the ground height network and the parameterized $\mathbf{M}$ estimation module in two steps. Because we find that it's hard for the network to converge when optimizing them simultaneously. In the first step, we train the ground height network and the learnable parameters of the camera/Radar height and coordinate deviations for 100 epochs. Then we use pre-trained weights and freeze the weights of the ground height network to train the parameterized $\mathbf{M}$ estimation

**Table 2: Ablation study on validation set in $Y = 150 - 330m$ range. We quantitatively evaluate the impact of the ground height network(GH), parameterized M estimation module(PE) and uncertainty estimation network(UE).**

| Base | GH | PE | UE | $\overline{AP}_{BEV}(\%) \uparrow$ | Loc. MAE$(m) \downarrow$ | | |
|---|---|---|---|---|---|---|---|
| | | | | | Lat. | Long. | Euc. |
| ✓ | | | | 63.9 | 0.142 | 1.217 | 1.239 |
| ✓ | ✓ | | | 66.3 | **0.128** | 1.024 | 1.045 |
| ✓ | ✓ | ✓ | | 66.5 | 0.148 | 1.011 | 1.039 |
| ✓ | ✓ | ✓ | ✓ | **68.5** | 0.151 | **0.951** | **0.982** |

LiDAR provides the ground truth data of multiple cars in this range.

**Table 3: Ablation study on validation set in $Y = 330 - 500m$ range.**

| Base | GH | PE | UE | $\overline{AP}_{BEV}(\%) \uparrow$ | Loc. MAE$(m) \downarrow$ | | |
|---|---|---|---|---|---|---|---|
| | | | | | Lat. | Long. | Euc. |
| ✓ | | | | 69.1 | 0.240 | 1.268 | 1.290 |
| ✓ | ✓ | | | 70.7 | 0.198 | 0.686 | 0.754 |
| ✓ | ✓ | ✓ | | 72.2 | **0.139** | 0.677 | 0.728 |
| ✓ | ✓ | ✓ | ✓ | **72.6** | 0.140 | **0.649** | **0.702** |

GPS provides the ground truth data of the experimentation vehicle in this range.

module and uncertainty estimation network for 70 epochs. We use the Adam optimizer in two training steps with batch size 8. The learning rate is initialized as 0.0001 for the first step and 0.001 for the second step, both are updated by the cosine annealing strategy.

## 4.2 Evaluation Results

We report the performance of our FARFusion V2 on the validation set for comparison with other methods and carefully investigate the effect of different modules. The results are evaluated using the average precision (AP) and mean absolute error (MAE) of location as illustrated in Sec. 4.1.

**Comparison results on the validation set.** We first compare the performance of different methods in $Y = 150 - 330m$ range in Tab. 1: (1) *Radar Only* (*i.e.,* obtaining a target's position using the Radar point location on the BEV plane), (2) *Image Only* (*i.e.,* obtaining a target's position by mapping its CBM point to BEV using the plane-assumed IPM), (3) *robustBEVfus* [1] (*i.e.,* the fusion of the previous two locations in BEV), (4) FARFusion [20] (*i.e.,* the target-level fusion of the Radar and CBM points on both 2D and BEV planes), and (5) our FARFusion V2. Among them, the first two are single modal while the latter three are fusion-based. The results show our FARFusion V2 has the highest $AP_{BEV}$ among all methods.

In addition, we have the following observations. Our FARFusion V2 has the highest $AP_{BEV}$ on the validation set, improving the average detection performance $AP_{BEV}$ by 5.4% than FARFusion and 28% than *robustBEVfus*. This demonstrates that 2D association is more effective in far-range scenes than BEV association used in *robustBEVfus* [1]. Moreover, our FARFusion V2 considers the impact of ground and sensor motions, achieving an improvement of 0.23$m$ in location accuracy compared to FARFusion.

Next, we perform a set of ablation studies in both $Y = 150 - 330m$ and $Y = 330 - 500m$ ranges and summarize the results in Tab. 2 and Tab. 3. We take the FARFusion as our baseline for comparison.

**Impact of the ground height network.** We first investigate the impact of the ground height network. As shown in Tab. 2, we obtain a 2.4% improvement in average $AP_{BEV}$ and a 0.194$m$ improvement in location accuracy when adding the ground height network (GH) in the target-level fusion. The ground height network can assign height values to the Radar data, thereby assisting the IPM process

**Table 4: Detection and location MAE results on validation set in $Y = 330 - 500m$ range.**

| Method | $\overline{AP}_{BEV}(\%) \uparrow$ | $AP_{BEV}(\%) \uparrow$ | | | | Loc. MAE$(m) \downarrow$ | | |
|---|---|---|---|---|---|---|---|---|
| | | $dis_{0.5}$ | $dis_{1.0}$ | $dis_{1.5}$ | $dis_{2.0}$ | Lat. | Long. | Euc. |
| *Radar Only* | 56.9 | 1.8 | 46.7 | **88.8** | **90.3** | 0.344 | 1.655 | 1.730 |
| *Image Only* | 1.8 | 0.2 | 1.3 | 2.7 | 3.3 | **0.067** | 1.617 | 1.618 |
| *robustBEVfus* [1] | 56.3 | 1.6 | 48.2 | 86.4 | 89.4 | 0.341 | 1.627 | 1.703 |
| FARFusion [20] | 69.1 | 18.8 | 73.0 | 91.1 | 93.8 | 0.240 | 1.268 | 1.290 |
| FARFusion *V2* | **74.5** | 44.8 | **79.4** | 86.3 | 87.6 | 0.172 | **0.712** | **0.771** |

GPS provides the ground truth data of the experimentation vehicles in this range. We filter out the detection results whose location errors $> 3.0m$ with GPS value for evaluation because other vehicles will influence the evaluation without ground truth.

by providing more accurate depth measurements. It achieves a more precise location accuracy of 1.045$m$. The experiment in the $Y = 330 - 500m$ range from Tab. 3 also demonstrates the effectiveness of the ground height network with 1.6% and 0.536$m$ improvements in $AP_{BEV}$ and location accuracy.

**Impact of the parameterized homography matrix estimation module.** We also show the results with the parameterized homography matrix estimation module (PE) in the third row in Tab. 2. However, we observe that the effect of this module in the $Y = 150 - 330m$ range is minimal with an improvement of only 0.2% in $AP_{BEV}$. This is because the sensor's motion has a greater influence at longer distances as illustrated in the introduction. Therefore, we also evaluate the effect of this module in the $Y = 330 - 500m$ range in Tab. 3. The results show adding this module yields a 1.5% improvement in $AP_{BEV}$.

**Impact of the uncertainty estimation network.** We finally evaluate the effect of the uncertainty estimation network (UE). In the last row in Tab. 2, we obtain a 2.0% improvement in $AP_{BEV}$. This demonstrates the depth values of the CBM points are more accurate when using the uncertainty-based weighting between the depths obtained via the Radar projection and those calculated by IPM than using only the depth obtained via the Radar projection directly. However, we also observe that the improvement of this module is minimal in $Y = 330 - 500m$ range in Tab. 3. This is because the IPM points have much higher location errors in this range. The uncertainty of the depth calculated by IPM is greater than the uncertainty of depth obtained by Radar projection. Therefore, the final weighted depth is highly dependent on the depth obtained through Radar projection, resulting in the performance being similar to that achieved without this uncertainty estimation network. We will analyse the uncertainty network in more detail in Sec. 4.3.

**Longer ranges with GPS as ground truth.** Finally, we extend the detection range to $Y = 330 - 500m$ to compare the performances of different methods in Tab. 4. Our FARFusion V2 still has the highest performance. We obtain a location accuracy of 0.771$m$ for the experimentation vehicles. This range has a high requirement for accurate view transformations. The ground height network and homography matrix estimation module play important roles in this range, as demonstrated in the preceding ablation studies.

## 4.3 Fine-grained Analysis

In this subsection, we delve into FARFusion V2 to study the ground height network and uncertainty estimation network qualitatively.

**What did the ground height network learn?** To investigate the outputs of the ground height network, we divide this area of $(-15m \leq X \leq 10m, 200m \leq Y \leq 700m)$ range in Radar CS into grids of uniform size of $2m \times 20m$. Then we take the grid points as

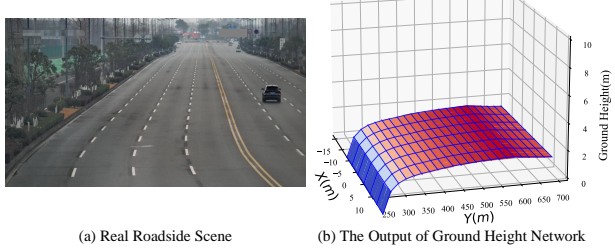

(a) Real Roadside Scene                    (b) The Output of Ground Height Network

**Figure 4: The comparison of the actual ground in figure (a) with the ground height network's output in figure (b).**

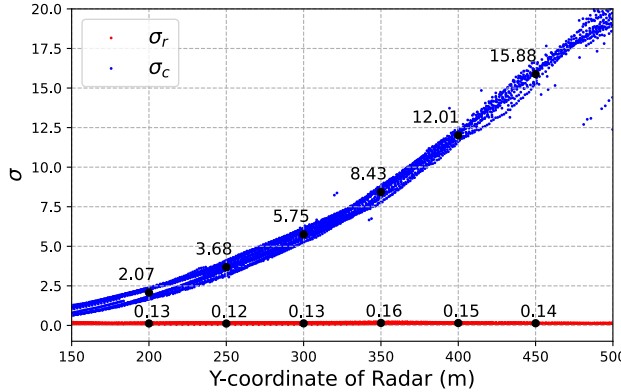

**Figure 5: The uncertainty of Radar projection depth and IPM-calculated depth with respect to the $Y$ coordinate in the BEV's Radar coordinate system.**

the input of the ground height network and plot the height values in Fig. 4(b). We observe that in the $Y = 300 - 600m$ range, there is an upward trend, while in the $Y > 600m$ range, it shows a downward trend in ground height. This trend is consistent with the ground's fluctuations in Fig. 4(a). However, we also observe that there is an unexpected sudden drop in the ground's height in the position of $Y = 200m$. We believe this is because we use fusion points, not ground truths, to train the network for ground height estimation. In practical scenarios, the fusion points are subject to inevitable detection noise with outliers, which causes the trained network to be imperfect. Even so, the overall surface contour of the output of the ground height network closely matches the actual ground.

**The analysis of the uncertainty estimation network.** The measurement accuracy of the sensor is highly relative to the detection range. The measurement noise usually becomes larger as the distance increases. To analyse the uncertainty predicted by the uncertainty estimation networks of Radar projection depth and IPM-calculated depth concerning the distance, we select matched Radar and 2D CBM points from 1000 frames randomly as the input of the network. Then we plot the uncertainty values $\sigma_r$ and $\sigma_c$ concerning the $Y$ coordinate of the BEV's Radar CS and mark the mean values in several locations in Fig. 5. We observe that the Radar projection depth [2] is more reliable than the IPM-calculated depth in the whole range. Meanwhile, the magnitude of $\sigma_c$ is significantly greater than that of $\sigma_r$ in the $Y > 300m$ range. This indicates that the camera's IPM performs poorly in $Y > 300m$ range, where the

---

[2]There is a downward trend of $\sigma_r$ in $Y > 350m$ range due to the use of fusion points as the ground truth for network training.

**Table 5: The impact of the coordinate deviation on the validation set in $Y = 150 - 330m$ range.**

| Base | $Coord_{dev}$ | $\overline{AP}_{\text{BEV}}(\%) \uparrow$ | Loc. MAE($m$) $\downarrow$ | | |
|------|---------------|-------------------------------------------|------|------|------|
| | | | Lat. | Long. | Euc. |
| ✓ | | 61.4 | **0.125** | 1.21 | 1.229 |
| | ✓ | 66.3 | 0.128 | **1.024** | **1.045** |

LiDAR provides the ground truth data of multiple cars in this range.

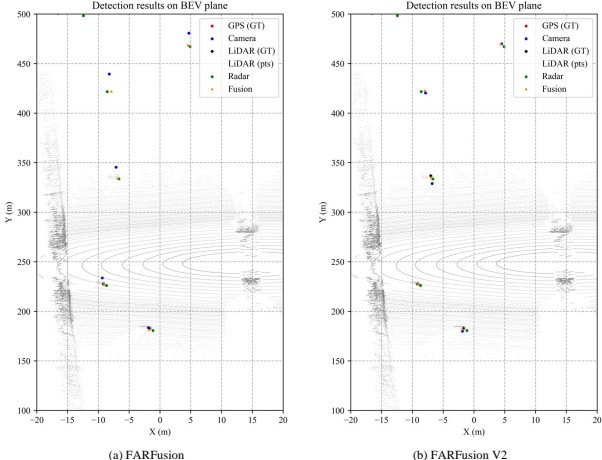

(a) FARFusion                    (b) FARFusion V2

**Figure 6: The qualitative results in Radar coordinate system of** FARFusion **in figure (a) and** FARFusion **V2 in figure (b).**

Radar projection depth becomes dominant in the depth fusion. This result is also consistent with the ablation studies in Sec. 4.2.

**Impact of the coordinate deviation in ground height network.** In Sec. 3.2, we have considered the translation deviations ($\Delta X, \Delta Y$) between the Radar CS and ground CS. Because the similar triangle relation ($\frac{X_c + \Delta X}{X_c' + \Delta X} = \frac{h_c - Z_g}{h_c}$) is only established with the ground CS while our BEV plane is in the Radar CS. We conduct an ablation study to analyse the effect of deviations in Tab. 5. The "base" represents we do not use translation deviations. The results show the performance is better with the translation deviations.

**The qualitative results in Radar CS of** FARFusion **and** FARFusion **V2.** Finally, we show the qualitative results on the BEV plane of FARFusion and FARFusion V2 in Fig. 6. We see that the FARFusion V2 has higher location accuracy, especially the IPM points highlighted in blue are adjusted to be closer to the accurate fusion points at larger ranges.

## 5    Conclusion

In this paper, to achieve high-accuracy location in far-range scenes, we exploit a geometry-based Radar and camera fusion method FARFusion V2 in target level on the ground for roadside 3D object detection. We find that ground height and sensor motion are important factors influencing the performance of roadside Radar and camera fusion. Therefore, we represent the ground height implicitly and employ a geometry-based transformation function for training. We also engage a parameterized homography matrix estimation module to refine the transformation parameters in real time. Additionally, we further improve the accuracy of depth values using an uncertainty-based depth fusion approach, supported by a mathematical theory derivation. With these designs, we achieve high-accuracy location performance for far-range object detection.

## Acknowledgments

This work was supported by the National Natural Science Foundation of China (No. 62332016).

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
