# OpenReview forum: "FARFusion V2: A Geometry-based Radar-Camera Fusion Method on the Ground for Roadside Far-Range 3D Object Detection"
_acmmm.org/ACMMM/2024/Conference — MM2024 Poster_

### Official Review · Reviewer_n6vu · 2024-05-24

**Rating:** 3
**Confidence:** 2

**Summary:**

This paper proposes FARFusion V2 to estimate ground height with arbitrary shapes and regress the transformation parameters to further refine the view transformations. It also includes an uncertainty-based depth fusion strategy to maximize the probability of obtaining the optimal depth value.

**Strengths:**

- The paper is largely clearly written and makes an easy read.
- The authors solve practical issues on top of FARFusion and show improvements over baselines.

**Limitations:**

- In both the abstract and the introduction, the authors do not explain the extent to which uneven terrain and sensor movement affect the precision of object detection. Without this information, we cannot determine if addressing these issues is necessary.

- Similarly, I am curious whether the 5.4% improvement in AP over FARFusion is substantial enough to justify this approach.

- Since the authors propose a real-time parameterized transformation parameters estimation module, it is essential to report its latency and the required latency. I’m wondering If the latency is small enough to process the real-time sensor motions.

- The proposed approach must demonstrate adaptability to various scenarios and conditions. Since the testbed only includes one type of scenario (urban expressway), it is necessary to apply the approach to other scenarios, such as roads with continuous changes in ground height. Additionally, how large can sensor motions be while still allowing the approach to provide accurate estimations?

**Suitability:**

2

---

### Official Review · Reviewer_r6BS · 2024-05-25

**Rating:** 4
**Confidence:** 2

**Summary:**

This paper aims to address the impact of uneven terrain and sensor movement on the Radar-camera fusion performance in the far-range 3D object detection task.
The main contributions include a geometric-based ground height learning method, a real-time transformation parameter estimation module, and an uncertainty-based depth fusion strategy.
Experiments on the roadside OWL dataset shows that it improves the object detection accuracy and the location accuracy in far-range.

**Strengths:**

The proposed method uses a geometric-based ground height learning network to reduce the influence of uneven terrain.
It also considers the sensor movement in windy weather and uses a real-time transformation parameter estimation module to improve the homography matrix.
Considering the observation noises from two sensors, the uncertainty-based depth fusion strategy is applied.

**Limitations:**

1. According to Figure 1, why does the sensor only move in one direction instead of moving in all three directions? Is it related to the fixing method of the sensor?
2. The pair fusion points (Xf, Yf) and 2D CBM points (Ucam, Vcam) are obtained based on the pre-calibrated M. What is the impact of the uneven terrain and sensor movement on the pair fusion points and the subsequent experiment?
3. Can the height change in the X direction also be reflected correctly as in the Y direction. The Figure 4 only reflects the height change in the Y direction.
4. For Table 2, it is better to add the results of PE, UE, GH+UE and PE+UE, to more clearly demonstrate the effectiveness and necessity of each module.

**Suitability:**

3

---

### Official Review · Reviewer_XGVX · 2024-05-26

**Rating:** 5
**Confidence:** 3

**Summary:**

This paper proposes FARFusion V2, which solves one problem of the existing FARFusion.
This method estimates the position and speed of a moving vehicle using fixed cameras and millimeter-wave radar, such as those used in road surveillance systems.
The proposed method achieves higher accuracy than when the shape and relative position of the road surface are fixed by geometrically considering the shift of the viewpoint due to the vibration of the system.

**Strengths:**

This is a practical method that addresses problems that arise in real-world environments, whereas many studies only pursue behavior in an ideal environment.
Furthermore, rather than leaving everything to machine learning, the proposed method attempts to solve the problem by considering geometric constraints.

**Limitations:**

-3D Object Detection-
Although it is described as an "object," it is thought to be specialized for "vehicles."

-machine learning-
Even though geometric constraints are taken into account, it is still insufficient to explain the advantages of going through machine learning.
It is necessary to explain whether the proposed method is effective in various environments, since the generalization performance problem arises as soon as ML is used.
(For example, does it work well when the road environment is not a wide road like the one shown in Figure 4, or when it is not a normal day but in rough weather with severe shaking?)

-OWL testbed-
This appears to be the authors' own dataset.
Will it not be made public?
If not, it cannot be used as a benchmark, so it is necessary to mention the behavior of the proposed method in areas other than a small number of ``urban expressways''.

-Figure 1(b)-
"Unexpected motion" means that only the angle from the sensor mount has changed, but in "real-time unexpected sensor motions still exist due to the windy weather," the rotation from the base of the pole is expected to be greater, so this explanation seems strange.

**Suitability:**

2

---

### Meta-Review · Area_Chair_KnFR · 2024-06-25

**Recommendation:** Accept (Poster)
**Confidence:** 5

**Metareview:**

The paper presents FARFusion V2, a geometry-based radar-camera fusion method designed to enhance far-range 3D object detection for roadside traffic surveillance. The proposed method addresses real-world challenges such as uneven terrain and sensor movement, demonstrating significant improvements in localization accuracy. The reviewers acknowledged the practicality and robustness of the approach, along with the extensive experiments validating its effectiveness. Concerns regarding the explanation of certain aspects and the need for additional evaluations were addressed in the authors' rebuttal. Therefore, I recommend accepting this submission.